# Early Diagnosis and Antibiotic Treatment Combined with Multicomponent Hemodynamic Support for Addressing a Severe Case of Lemierre’s Syndrome

**DOI:** 10.3390/antibiotics10121526

**Published:** 2021-12-14

**Authors:** Andreaserena Recchia, Marco Cascella, Sabrina Altamura, Felice Borrelli, Nazario De Nittis, Elisabetta Dibenedetto, Maria Labonia, Giovanna Pavone, Alfredo Del Gaudio

**Affiliations:** 1Anesthesia and Intensive Care 2, IRCCS “Casa Sollievo della Sofferenza”, 71013 San Giovanni Rotondo, Italy; sabrina.altamura@icloud.com (S.A.); f.borrelli@operapadrepio.it (F.B.); n.denittis@hotmail.it (N.D.N.); dib.elisabetta@gmail.com (E.D.); g.pavone@yahoo.it (G.P.); a.delgaudio@operapadrepio.it (A.D.G.); 2Anesthesia and Pain Medicine Istituto Nazionale Tumori-IRCCS, Fondazione G. Pascale, 80100 Napoli, Italy; 3Microbiology and Virology, IRCCS “Casa Sollievo della Sofferenza”, 71013 San Giovanni Rotondo, Italy; m.labonia@operapadrepio.it

**Keywords:** Lemierre’s syndrome, necrobacillosis, post-anginal septicemia, septic shock, norepinephrine, argipressin

## Abstract

A 20-year-old man was admitted to the intensive care unit for septic shock due to Lemierre’s syndrome. It is a rare syndrome that manifests as an upper respiratory infection, although systemic involvement, severe coagulopathy, and multi-organ failure can dangerously complicate the clinical picture. In this syndrome, sepsis-related neuroendocrine dysregulation and microcirculation impairment can have a rapid deleterious progression. Consequently, proper diagnosis, early source control, and appropriate antibiotics administration are mandatory to improve the prognosis. The intensive treatment is aimed at limiting organ damage through hemodynamic optimization. Remarkably, in septic shock due to Lemierre’s syndrome, hemodynamic optimization can be achieved through the synergic effect of norepinephrine, argipressin, and hydrocortisone.

## 1. Introduction

Lemierre’s syndrome, also known as post-anginal septicemia or necrobacillosis, is a rare clinical condition with high morbidity, frequent evolution to septic shock, high intensive care unit (ICU) admission rate, and prolonged hospital length of stay [1]. It usually affects previously healthy adolescents and young adults with an almost double incidence in males compared to females [2,3]. The incidence is about 15 cases per million per year in individuals aged 15–24 years old [4]. In the pre-antibiotic era, the syndrome was a common infectious disease with fatal outcomes in most cases. Subsequently, due to the use of antibiotics for upper respiratory tract infections, its incidence greatly reduced, and this severe clinical condition became a “forgotten syndrome” [5,6]. In recent years, the rise of the cases is probably due to the augmented awareness of the syndrome, the careful prescription of antibiotics for the management of streptococcal pharyngitis, and a growing antibiotic resistance expressed by the bacterial pathogen (e.g., towards macrolides) [1,6,7]. Although several bacteria such as *Fusobacterium nucleatum*, streptococci, staphylococci, *Klebsiella pneumonia*, and others can cause this syndrome, the pathogen most frequently involved is *Fusobacterium necrophorum*. It is a Gram-negative pleomorphic anaerobic bacterium found in the normal bacterial flora of the oral cavity but also in the gastrointestinal and female genital tract. Polymicrobial bacteremia with a combination of Gram-positive and Gram-negative anaerobes can also be found [8].

About the pathogenesis of the syndrome, *Fusobacterium necrophorum* produces a lipopolysaccharide endotoxin and several exotoxins. These latter are proteolytic enzymes that usually induce a systemic inflammatory response. Among the exotoxins, hemolysin creates an anaerobic environment by lysing erythrocytes and limiting the transport of oxygen to the site of primary infection. It produces an environment favorable to the growth of the pathogen, favoring the invasion of regional veins [3]. This process can activate the intrinsic pathway of coagulation and, finally, disseminated intravascular coagulopathy [5]. When jugular vein thrombosis occurs, the bacteremia can lead septic emboli to different organs, such as lungs, liver, spleen, kidneys, joint, bone, and the central nervous system [6]. Rarely, this cascade of events can finally culminate in septic shock and multi-organ failure.

Here, the authors present a severe case of Lemierre’s syndrome with special attention to the diagnosis, proper antibiotic therapy, and appropriate management of sepsis-related neuroendocrine dysregulation and microcirculation impairment.

## 2. Case Presentation

### 2.1. History and Hospital Admission

A 20-year-old Caucasian male (1.75 m tall and 76 kg (BMI 24.8)), was admitted to the medical department for persistent hyperpyrexia, severe sore throat, dyspnea, and impaired consciousness with stupor. Persistent symptoms started at home 4 days before and he assumed clarithromycin as empiric antibiotic therapy. The physical examination showed jaundice, dry mucous membranes, pharyngeal hyperemia in the tonsillar region and soft palate, and left laterocervical lymphadenopathy. He was tachypneic (respiratory rate of 30 breaths per minute) and the peripheral oxygen saturation (SpO_2_) in room air was 92%. The abdominal palpation revealed hepatosplenomegaly. The laboratory tests showed a white blood count (WBC) of 8000 cells/mcL with 74% neutrophils, thrombocytopenia (platelet count of 31,000/mcL), total bilirubin 5.8 mg/dL, C-Reactive Protein (CRP) 43 mg/L, creatinine 0.9 mg/dL, AST 150 UI/L, ALT 79 UI/L. The nasopharyngeal swab testing for SARS-CoV-2 was negative (RT-PCR). Blood cultures were carried out upon admission and a full-body computer tomography (CT) was performed on the second day of hospitalization. The CT showed ground glass bilateral pulmonary alterations, pericardial effusion, mediastinal lymphadenopathy, and hepatosplenomegaly (Figure 1).

The neck CT scan with intravenous contrast evidenced a 5.4 cm retropharyngeal abscess with associated thrombosis of the left anterior jugular vein (Figure 2).

On the second day of hospitalization, the microbiology laboratory communicated the early identification of *Fusobacterium necrophorum* grown in blood cultures by MALDI-TOF (Matrix Assisted Laser Desorption Ionization Time-of-Flight) spectrometry -Vitek ^®^MS Blood cultures performed at admission and on the second day and collected in standard anaerobic blood culture bottles were positive (Figure 3).

The association of retropharyngeal abscess with thrombosis of the anterior jugular and involvement of other organs systemically and the microbiology led to the diagnosis of Lemierre’s syndrome. Despite empirical antibiotic therapy with piperacillin/tazobactam (18 g/day, continuous infusion), there was a rapid worsening of the clinical conditions with further impairment of the respiratory failure and severe hemodynamic alterations (mean arterial pressure (MAP) constantly <65 mmHg). This picture suggested a progression towards a septic shock status. Consequently, the patient was referred to the Intensive Care Unit (ICU).

### 2.2. Intensive Care Unit Management

A multimodal approach was the core of the ICU management of Lemierre’s septic syndrome. It included source control, antibiotic therapy, anticoagulant therapy, hemodynamic support, and continuous renal replacement therapy (CRRT) (Figure 4).

The source control was planned in a sterile environment. Under general anesthesia, the patient underwent an explorative puncture of the retropharyngeal abscess and surgical tracheostomy for airway protection. Unfortunately, although the procedure revealed purulent material, surgeons were unable to collect suitable samples for microbiological testing.

At ICU admission, the patient showed a clinical and radiological picture of acute lung injury and the need for deep sedation (propofol, remifentanil, and clonidine), and mechanical ventilation through protective ventilation (tidal volume 6 mL/kg and driving pressure < 15 cmH_2_O) and high FiO_2_ (80%). The sequential organ failure assessment (SOFA) score was 12, predicting a high mortality rate (50%) [9].

About antibiotic therapy, metronidazole (500 mg q8hr, intravenously) was added to piperacillin/tazobactam (started empirically at hospital admission). This antibiotic therapy was based on anecdotal clinical evidence because, due to the lack of cards for antimicrobial susceptibility testing of anaerobes in the laboratory, the antibiogram was not performed.

Hemodynamic support was based on the synergic association of norepinephrine and the non-adrenergic vasopressor argipressin (also known as arginine-vasopressin or anti-diuretic hormone) (EMPRESSIN^®^ AOP Orphan Pharmaceutical Group) starting with norepinephrine at 0.3 mcg/kg/min and argipressin at 0.03 IU/min to maintain a MAP > 65 mmHg and lactic acid < 2 mmol/L. On day 2, the drug synergism allowed the rapid reduction of the norepinephrine and argipressin dosages (0.1 mcg/kg/min and 0.025 IU/min, respectively) (Figure 5).

On the same day (day 2), there was an enhancement in the clinical conditions, and the biomarkers of inflammation and organ failure improved (Table 1).

Despite the adequate MAP (>65 mmHg), from the 3rd to the 6th day of ICU admission, there was a new worsening of the clinical status. It featured anemia (worse value 6.8 g/dL) without evidence of a hemorrhagic source. The hemodynamic monitoring (Edwards EV1000™ device) showed a hyperdynamic septic state (Cardiac Output 15 L/min, Systemic Vascular Resistance 300 dynes/seconds/cm). Due to the worsening of the sepsis-related organ damage, on the 8th day, hydrocortisone (200 mg/day) was added to argipressin and norepinephrine. This approach led to a paramount hemodynamic improvement. From the 8th day of hospitalization, there was a progressive reduction of the dose of both vasoconstrictors until the suspension of norepinephrine, on day 9, and argipressin, on day 12 of ICU stay. Moreover, there was an important decrease in the Sequential Organ Failure Assessment (SOFA) score (from 12 on ICU admission to 7 on day 9).

The worsening of the clinical status and the anemia were associated with bilirubin and cytokines rise. The inflammatory response was treated by 4 cycles of continuous hemofiltration (from 4th to 7th day). At the end of the procedure, there was an unexpected rise in D-dimers and in bilirubin values, up to 15.3 mg/dL (10th day), as well as hyperpyrexia. Nevertheless, these findings improved within 24 h.

Heparin therapy was administered throughout the entire stay according to thrombocytopenia (enoxaparin 6000 U/day at the beginning and 4000 U twice a day after normalization of the platelet count) monitoring D-dimer values and avoiding intravascular coagulopathy.

On the 9th day, a chest tube was placed for the drainage of lung pleural effusion and a new CT was carried out. It showed a persistence of the known injuries of the lungs, liver, and spleen despite the paramount improvements of laboratory and clinical values, as well as the normalization of hemodynamic parameters at monitoring. The patient was discharged to the ward on the 17th day to continue the antibiotic therapy and the rehabilitation program. After the discharge to the rehabilitation care unit, the young man was in good health condition. At the first follow up, performed 15 days after ICU admission, the CT scan showed better lungs ventilation, absence of retropharyngeal abscess, laterocervical lymphadenopathy, and internal jugular thrombosis. Hepatosplenomegaly and some cavitary lungs lesions were found. Another thoracic CT scan was performed after 3 months. It demonstrated an improvement of both the lung lesions and hepatosplenomegaly.

## 3. Discussion

Lemierre’s syndrome is a serious infectious disease with septic thrombophlebitis of the internal jugular vein that can hesitate in a clinical picture of multi-organ impairment. Consequently, early diagnosis, source control, immediate antibiotic therapy, and optimized hemodynamic treatment are crucial to improving the patient’s prognosis. This case report shows that the rapid worsening of the clinical conditions and laboratory findings was probably due to systemic emboli and the production of bacterial exotoxins (hemolysin, heparinase, and other proteolytic enzymes) with a hyperdynamic state and hemolysis. In the literature, severe clinical presentations with septic shock are described as case reports [10,11]. The peculiarity of this case is the early diagnosis, the proper antibiotic therapy, and the drug combination used against the pathogenetic storm triggered by the infection. Multi-professional management was fundamental for achieving a positive outcome.

The CT finding of retropharyngeal abscess associated with anterior jugular vein thrombosis was helpful to the clinical diagnosis of the disease. Moreover, the spectrometry allowed the quick identification (minutes) of *Fusobacterium necrophorum*.

A set of clinical and instrumental data (contrasted CT scan of the neck) must lead to suspicion of the disease. It is mandatory to start the proper (empiric) therapy as soon as possible before waiting for the laboratory confirmation. This approach is of paramount importance, especially when rapid microbiological diagnosis is not available. According to Wright et al. [12], the key elements for the diagnosis are pharyngotonsillitis:Without resolution in less than a week;With unusual lateral cervical pain, dysphagia, and swelling as well as trismus, unilateral (or bilateral) anterior cervical lymphadenopathy, and induration at the angle of the mandible;With diffuse abscesses (imaging);Followed by systemic (e.g., fever, malaise, confusion, stupor) and/or respiratory findings such as dyspnea or tachypnea;Succeeded by a progressive worsening of the clinical picture with hemodynamic and laboratory problems suggestive for sepsis.

Furthermore, pulmonary involvement, which is produced by metastasis of septic emboli, is frequently observed during Lemierre’s syndrome [10,11,12,13]. However, in our case, the patient was admitted to the ICU with acute respiratory distress syndrome. Subsequently, he developed a pleural effusion. Probably, the severe respiratory damage had a multifactorial origin as the infectious insult (septic emboli) was associated with the damage due to hypoperfusion. In the context of multimodal management, protective ventilation and improvement of oxygen release were of fundamental importance.

Source control is an essential element in the management of sepsis and should be performed soon after the diagnosis is established in all patients [14]. Early open surgical drainage remains the most appropriate method of treating a deep neck abscess. In adult patients, there was a significant increase in abscess-specific morbidity and mortality with delay in incision and drainage [15]. The treatment of deep neck infection consists of securing the airways and surgical drainage of the abscesses. Moreover, the therapeutic use of needle aspiration has been suggested in selective cases [16]. It is quick to perform and less invasive, and there is no evidence that this technique is less effective than surgical drainage [17]. Regardless of the technique used, in Lemierre’s syndrome source control is mandatory. In a recent analysis of 218 patients with deep neck infection with or without mediastinal involvement, the authors showed that immediate surgical drainage, broad empiric antibiotic therapy, surgical revision, antiseptic wound lavage, and early tracheostomy are the cornerstones of therapy [18]. Early tracheostomy is important to facilitate surgical revision and reduce sedation time and artificial ventilation duration [19,20].

Delay in the initiation of appropriate antibiotic therapy has been recognized as a risk factor for mortality, with a linear increase in the risk for each hour of delay in antibiotic administration [21]. *Fusobacterium necrophorum* is intrinsically resistant to macrolides, fluoroquinolones, tetracyclines, and aminoglycosides. In England and Wales, from 1990–2000, of 208 isolations of *Fusobacterium necrophorum*, 1% were resistant to tetracycline, 2% to penicillin, and 15% showed resistance or reduced sensitivity to erythromycin [22]. Overall, in an old study from Austria, most isolates (*n* = 36) were sensitive to all tested antibiotics (amoxicillin/clavulanic acid, imipenem, clindamycin, and metronidazole); only 2 of the 35 tested isolates showed resistance, one isolate against penicillin G and the second against metronidazole [23]. In Denmark, 357 isolates of *Fusobacterium necrophorum* were consecutively collected over a 3-year period. The MIC (mg/L) was determined only for 40 samples. The MIC90 was 0.047 for penicillin, 0.047 for clindamycin, 0.25 for metronidazole, 0.38 for cefuroxime, >32 for imipenem, 0.012 for meropenem, and 2 for erythromycin. All 357 isolates were susceptible to penicillin and metronidazole. However, after the study period, a penicillin-resistant strain of *F. necrophorum* was isolated from a throat swab [24]. From other literature data, it emerged that normally, *F. necrophorum* is susceptible to penicillin, cephalosporins, metronidazole, clindamycin, tetracyclines, and chloramphenicol. Moreover, β-lactamase-producing strains of *F. necrophorum* have only very rarely been reported. On the other hand, other Fusobacterium species, and concomitant bacteria, may produce β-lactamase and make penicillin therapy insufficient [25,26]. These data indicate that β-lactam/β-lactamase inhibitor combinations, metronidazole, and clindamycin (especially in cases of penicillin allergy) are the drug of choice for the treatment of *F. necrophorum* infections. Since polymicrobial bacteremia (*F. necrophorum* plus other oral flora) is often found, it was suggested that monotherapy with metronidazole should be avoided [8]. Remarkably, piperacillin/tazobactam is considered a weak inducer of beta-lactamases such as AmpC and, therefore, it is the most used empiric beta-lactam in Italy, not for the anti-Pseudomonas activity [27,28]. Likewise, in Italy, to avoid a stronger selection of MDR microorganism in the microbiota of treated patients, piperacillin/tazobactam is used for its activity against ESBL Gram-negative bacteria according to a carbapenem-sparing strategy. Different from countries of northern Europe, Italy is an ESBL-country, piperacillin/tazobactam is preferred to ampicillin/sulbactam, although the anti-anaerobic activity is similar.

Unfortunately, due to the lack of cards for antibiotic susceptibility of anaerobic bacteria, the testing was not performed. Thus, we referred to literature data and clinical response. The infectious diseases clinician suggested the therapeutic choice of continuing the administration of piperacillin/tazobactam in association with metronidazole after the bacterial identification. Although the in vitro sensitivities were not performed, and to date, no specific guidelines are available to recommend the optimal antibiotic regimen, the strategy we followed provided a combination of the most prescribed antibiotics [8]. In line with what was indicated in the literature, the treatment lasted for about 3 weeks [3,10].

The use of anticoagulants in Lemierre’s syndrome is controversial [1,2,3,6]. Although less severe clinical presentations of the disease without evidence of massive thrombotic phenomena usually do not require anticoagulation, it may be recommended when the disease features extensive clotting, multi-organ involvement, and a progressive worsening of the clinical picture [6].

About hemodynamic treatment, an interesting therapeutic choice was the administration of triple synergic therapy with the association of argipressin, norepinephrine, and hydrocortisone. This approach is aimed at limiting the inadequate oxygen delivery and the development of the anaerobic metabolism. Although norepinephrine is usually used as a first-line agent to support hemodynamic [29], at high dosage, it is associated with significant collateral effects such as renal failure, decreased myocardial perfusion, and an increase of pulmonary vascular resistance. Vasodilatory shock with vasoplegia and diminished responsiveness to vasopressor therapy is the final common pathway of all forms of severe shock [30]. In severe septic shock, many mechanisms can cause resistance to norepinephrine treatment such as ATP-sensitive potassium channel alteration [31], nitric oxide mediate vasodilatation, endothelium-derived hyperpolarizing factors, vasoconstrictors receptors’ down-regulation, and hyposensitivity [32]. Furthermore, early septic shock includes a high endogenous circulating concentration of catecholamines and a dysregulated autonomic and neuroendocrine response with relative adrenal insufficiency and vasopressin insufficiency [33]. Vasopressin, also known as argipressin, arginine vasopressin (AVP), or anti-diuretic hormone (ADH), is a cyclic nonapeptide hormone with a vital hemodynamic effect of maintaining the vascular muscular tone. It is synthesized in the magnocellular neurons of the paraventricular and supraoptic nuclei of the hypothalamus and stored as prohormone granules in the pars nervosa of the posterior pituitary gland in storage granules. Notably, only 10–20% of the stored hormone can be rapidly released, explaining the biphasic response observed in septic shock [34]. During shock, the argipressin trend is characterized by an early peak, and a late drop to basal levels as shock becomes established, which is believed to worsen vasoplegia [35,36]. The Vasopressin and Septic Shock Trial (VASST) [37], a large multicenter randomized controlled investigation, addressed the synergic action between norepinephrine and vasopressin, comparing the use of norepinephrine alone with that of norepinephrine and low-dose vasopressin (0.01–0.03 IU/min) in 778 patients with septic shock. No significant difference in the primary endpoint (i.e., all-cause mortality at 28 days) was found between the two groups, although in a subgroup of patients with less severe septic shock, there was decreased mortality after vasopressin treatment. Nevertheless, a post hoc analysis of the trial data indicated a trend toward both improved renal function with vasopressin and a survival benefit when corticosteroids were co-administered [38]. Patients who received vasopressin did not show differences in total norepinephrine decreasing doses in the two groups with or without corticosteroid treatment, but the rate of survival was greater when vasopressin and corticosteroids were associated. Furthermore, the plasma concentrations of vasopressin were significantly higher at 6 and 24 h in patients treated with vasopressin and corticosteroids.

The effect of corticosteroids on septic shock is debated. Annane et al. [39] ensured that hydrocortisone or fludrocortisone decreased the mortality of patients with septic shock, whereas the CORTICUS study demonstrated that hydrocortisone did not change the mortality of these patients [40]. Moreover, the VANISH trial, which included a cohort study on the association of hydrocortisone and early vasopressin therapy, found no differences in mortality, but patients treated with hydrocortisone and vasopressin had a more rapid decrease in vasopressor requirements [41]. Furthermore, Schurr et al. [42] analyzed the therapeutic rationale of neuroendocrine derangement in early septic shock. They showed that in the initial septic phase, there is a hypersecretion of cortisol and vasopressin due to the number of cytokines that enter the hypothalamic-pituitary-adrenal axis. The release of ACTH from the anterior pituitary leads to corticosteroid expression in the adrenal cortex. However, in a subset of patients, this response may be inadequate. After an early phase of septic shock, there is a downregulation of ACTH synthesis not compensated by corticotropin-releasing hormone, or vasopressin, with a subsequent relative adrenal insufficiency. The synergism of hydrocortisone and vasopressin is due to different complex mechanisms [43]. Notably, vasopressin binds to V3 receptors located in the anterior pituitary, may increase ACTH production and secretion, and can directly stimulate adrenal glucocorticoid production. Norepinephrine inhibits the anti-diuretic effect of vasopressin in the kidney but requires cortisol. As discussed by Gordon et al. [43], corticosteroids can increase vasopressin mRNA, do not change vasopressin level, delay the vasopressor release, and can reverse the downregulation of vasopressin 1A receptors.

In this case of septic shock due to Lemierre’s syndrome, the synergic effect of argipressin, norepinephrine, and hydrocortisone has probably allowed the optimization of the oxygen transport, maintaining an average arterial pressure above 65 mmHg and lactic acid < 2 mmol/L. On the first day, the early addition of argipressin to norepinephrine led to a rapid decrease in the norepinephrine dosage with reduced values of lactic acid, although the high PCT and IL-6 levels. The worsening of the laboratory values at the 4th day suggested a progressive organ damage concomitant with the release of bacterial proteolytic enzymes; nevertheless, this damage was limited and did not lead to a further increase of lactates (Figure 6).

## 4. Conclusions

Lemierre’s syndrome can present as a severe clinical condition requiring complex multiprofessional management. The clinical picture of septic shock is the effect of a rapidly evolving pathogenetic process that culminates in vasoplegia, irreversible hypotension, and severe oxygen delivery impairment. Remarkably, vasoplegia is considered as a key factor responsible for the death of patients with septic shock. Early diagnosis, source control, initiation of adequate antibiotic therapy, careful anticoagulant therapy, and hemodynamic optimization are of fundamental importance to improve the prognosis of these patients. It can be suggested that in the treatment of septic shock due to Lemierre’s syndrome, the cornerstone can be the rapid correction of the neuroendocrine dysregulation through combined strategies aimed at the maximization of the oxygen delivery. In these fragile patients, the neuroendocrine dysregulation with vasoplegia can be successfully supported through the administration of triple synergic therapy with norepinephrine, vasopressin, and hydrocortisone. The effectiveness of the treatment can be demonstrated by the amelioration of different inflammatory markers.

## Figures and Tables

**Figure 1 antibiotics-10-01526-f001:**
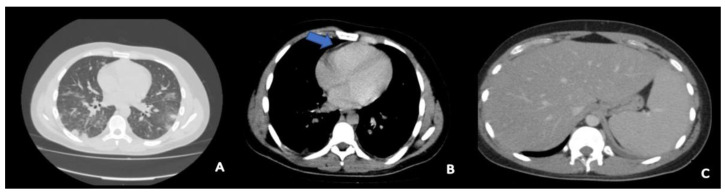
Computed tomography scan at admission to the intensive care unit. It showed bilateral ground glass pulmonary alterations (**A**), small pericardial effusion (blue arrow in **B**), hepatosplenomegaly (**C**).

**Figure 2 antibiotics-10-01526-f002:**
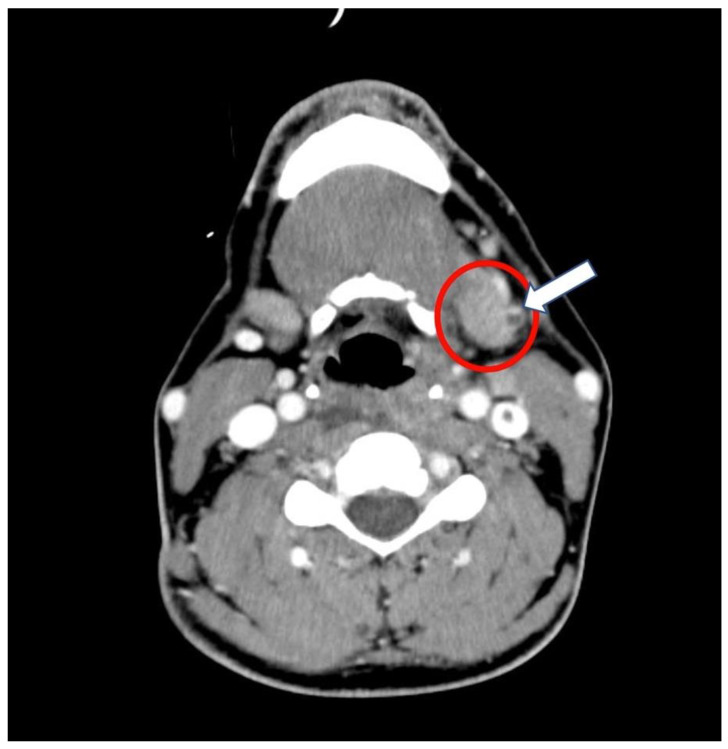
Computed tomography and head scan with intravenous contrast at admission to the intensive care unit. Retropharyngeal abscess (red circle) associated with anterior jugular thrombosis (arrow).

**Figure 3 antibiotics-10-01526-f003:**
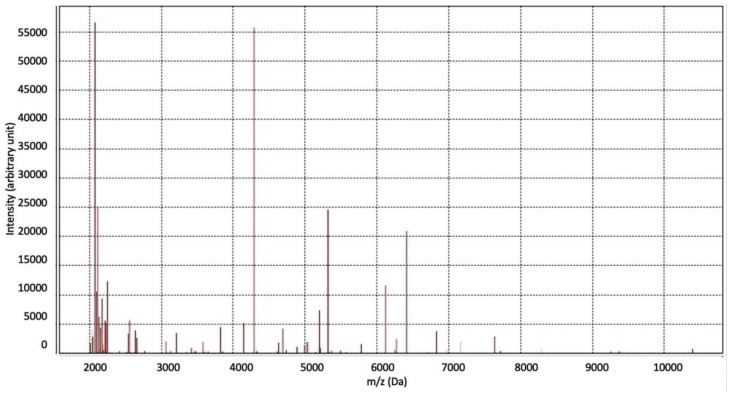
Spectrometry of MALDI-TOF (Matrix Assisted Laser Desorption Ionization Time-of-Flight) (Vitek ^®^MS) allowed the quick identification (minutes) of *Fusobacterium necrophorum* in blood culture at 48 h. This approach of mass spectrometry analyzes proteins (mainly ribosomal) of microorganisms in the mass range. The proteins are ionized into charged molecules to measure the mass to charge (*m*/*z*) ratio. Ions are accelerated and separate each other on the basis of their *m*/*z* ratio. A characteristic mass spectrum is generated with peaks that are specific to types of microorganisms. The relative intensities of the ions (a.u, arbitrary unit) are shown on the *y* axis, and the mass to charge ratio in the *x*-axis (in Da).

**Figure 4 antibiotics-10-01526-f004:**
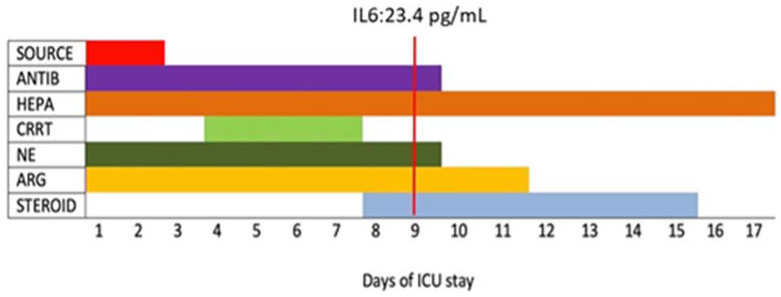
Timeline of therapeutic approaches from the intensive care unit admission to discharge. At day 9, there was a paramount reduction of inflammatory markers. Abbreviations: SOURCE: Source Control; HEPA: heparin treatment; ANTIB: antibiotics; CRRT: continuous renal replacement therapy; NE: norepinephrine; ARG: argipressin; STEROID: steroid therapy; IL6: interleukin 6.

**Figure 5 antibiotics-10-01526-f005:**
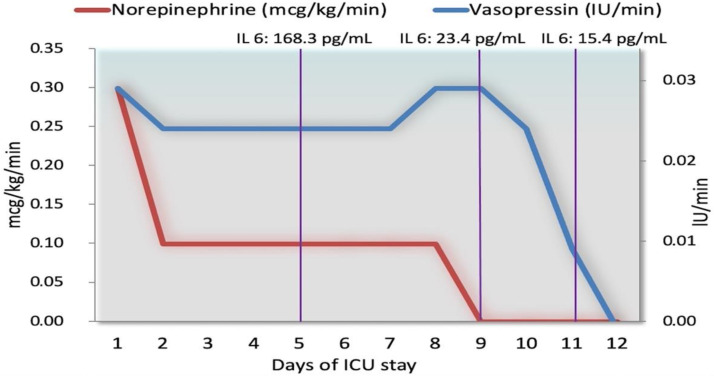
Norepinephrine and vasopressin infusion during the days of hospitalization. The early addition of vasopressin (0.03 IU/min) to norepinephrine (0.3 mcg/kg/min) led to a rapid decrease in the norepinephrine dosage at 0.1 mcg/kg/min and vasopressin steady infusion at 0.01 IU/min. The stability of norepinephrine infusion was maintained over the entire period of hospitalization until the suspension of norepinephrine (day 9), and vasopressin (day 12). There was a concomitant decrease in interleukin 6 (IL-6).

**Figure 6 antibiotics-10-01526-f006:**
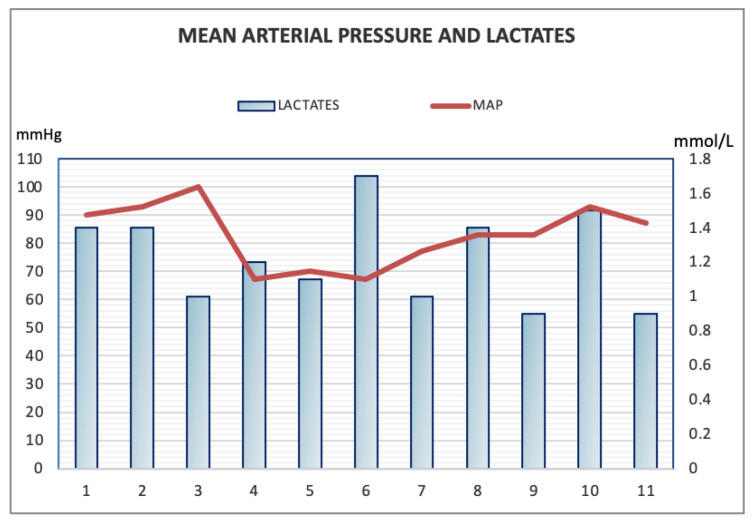
Trend of mean arterial pressure and lactates during the intensive care unit stay. Acid Lactic values were never above 1.6 mmol/L and mean arterial pressure was maintained above 65 mmHg.

**Table 1 antibiotics-10-01526-t001:** Laboratory, clinical data, and vasoconstrictors doses during the intensive care unit stay.

	Day 1	Day 2	Day 4 to 6 ^1^	Day 9	Day 11	Day 17
**Laboratory**						
WBC (10^3^/mcL)	12.8	12.45	16.49	11.86	6.06	6.66
Platelets (10^3^/mcL)	62	94	86	312	318	266
Hemoglobin (g/dL)	12.3	10.6	6.8	8.2	8.1	9.1
Procalcitonin (ng/mL)	100.7	56.6	13.3	2.09	1.22	0.2
IL-6 (pg/mL)	No data	No data	168.3	23.4	15.4	No data
Fibrinogen (mg/dL)	811	636	519	613	504	422
D-dimer (ng/mL)	2514	No data	4177	2266	5574	No data
Total bilirubin (mg/dL)	7.5	5.5	6.9	12.5	14.5	3.9
Creatinine (mg/dL)	0.8	0.8	0.8	0,7	0.6	0.5
**Clinical data**						
Temperature (°C)	38.3	38.8	39.3	39.7	38	37.2
PaO_2_/FiO_2_ (FiO_2_%)	298 (80%)	259 (55%)	77 (80%)	134 (0.65)	248 (0.5)	355 (0.3) ^2^
Lactic acid (mmol/L)	1.4	1.4	1.8	0.9	1.4	1
Blood Pressure (mean) (mmHg)	130/70 (90)	140/70 (93)	100/50 (95)	130/60 (83)	120/70 (87)	120/60 (80)
Heart rate (rpm)	100	100	95	100	60	80
Cardiac Output (L/min)	No data	No data	15	No data	8.7	No data
SVR ^1^ (dynes/seconds/cm) SOFA SCORE	No data 12	No data 10	300 13	No data 7	917 6	No data 3
**Vasoconstrictors doses**						
Argipressin (IU/min)	0.03	0.025	0.025	0.03	0.01	No drug
Norepinephrine (mcg/kg/min)	0.3	0.1	0.1	No drug	No drug	No drug

^1^ Worse values are reported. ^2^ In spontaneous breathing; Abbreviations: WBC: White blood cells; IL-6: interleukin 6; SVR: Systemic vascular resistance; SOFA SCORE: Sequential Organ Failure Assessment.

## Data Availability

The data presented in this manuscript are available on request from the corresponding author and approval by the consented patient involved in this report. The data are not publicly available to protect patient privacy and identity.

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
