# Peer review of "Early Diagnosis and Antibiotic Treatment Combined with Multicomponent Hemodynamic Support for Addressing a Severe Case of Lemierre’s Syndrome"

_antibiotics, 2021, doi:10.3390/antibiotics10121526_

Round 1
Reviewer 1 Report
This is a well written paper on a fairly uncommon syndrome Lomira syndrome and prevents a well described case report of extensive anti-microbial and I see you support of a patient with this rare syndrome and is of utility to clinicians treating patients with this rare syndrome
Author Response
COMMENT: This is a well written paper on a fairly uncommon Lemierre’s syndrome and prevents a well described case report of extensive anti-microbial and I see you support of a patient with this rare syndrome and is of utility to clinicians treating patients with this rare syndrome.
RESPONSE: thanks for these words of appreciation. I hope that after the revision this work will further meet your expectations.
Reviewer 2 Report
The paper was a case review of a successful treatment for a rare disease.
- Please review for appropriate grammar. English editing is needed in this manuscript. The authors should consider sourcing out the manuscript to a native English speaker to help with this matter
- The history of this patient is missing. What was the chief complaint?
- In line 80, the sentence means that the CT showed ground glass bilateral pulmonary alterations, pleural and pericardial effusion, mediastinal lymphadenopathy, and hepatosplenome, however, Figure 1 does not show all these findings. Do you have other CT scans?
- In Figure 2, please identify retropharyngeal abscess with another arrow.
- In line 100, was antibiotic sensitivity test not done with the cultured F. necrophorum?
- In line 129, can you specify source control? How was it done?
- Can you describe the op findings in surgical drainage of the retropharyngeal abscess? Any pus? Culture of pus?
- In figure 5, can you point the paramount reduction of inflammatory markers, as in Figure 4?
- Did you check the SOFA scores daily? If so, can you put them in Table 1?
- Was follow up CT not done? Findings?
- The discussion is rather long.
- Rather than describing each treatment options for Lemierre's syndrome, it is better to present treatment combinations (or cases) that have been reported to successfully treat Lemierre's syndrome. Is this the first report to treat Lemierre's syndrome with your treatment combination in the literature?
- The conclusion is too vague, it should be more specific that reflects your specific study results. Can you only present how you treated this patient? Was anticoagulant therapy done in this patient?
Author Response
COMMENT: Please review for appropriate grammar. English editing is needed in this manuscript. The authors should consider sourcing out the manuscript to a native English speaker to help with this matter
RESPONSE: Thanks for this suggestion. The test has been submitted to the attention of a native speaker.
COMMENT: The history of this patient is missing. What was the chief complaint?
RESPONSE: We have further evaluated the clinical documentation and new emerging findings (e.g., laboratory data) were added. No further data to be reported emerged.
COMMENT: In line 80, the sentence means that the CT showed ground glass bilateral pulmonary alterations, pleural and pericardial effusion, mediastinal lymphadenopathy, and hepatosplenomegaly, however, Figure 1 does not show all these findings. Do you have other CT scans?
RESPONSE: Yes, we performed a total body CT. Other images were included (e.g., pericardial effusion, hepatosplenomegaly). (see the revised Figure 1).
COMMENT: In Figure 2, please identify retropharyngeal abscess with another arrow
RESPONSE: Thanks for the suggestion, we revised the picture as required.
COMMENT: In line 100, was antibiotic sensitivity test not done with the cultured F. necrophorum?
RESPONSE: Unfortunately, the antibiotic sensitivity test was not performed. This gap was explained in section 2.2. In our hospital, cards for testing the susceptibility to antibiotics for anaerobic bacteria are not available. Thus, we referred to literature data (references from 22 to 26) for the antibiotic therapy as in the reference list.
COMMENT: In line 129, can you specify source control? How was it done? Can you describe the op findings in surgical drainage of the retropharyngeal abscess? Any pus? Culture of pus?
RESPONSE: According to your suggestion, this point was better explained, and the issue of the source control was addressed before explaining the clinical and radiological findings. The source control was planned in a sterile environment. Under general anesthesia, the patient underwent to an explorative puncture of the retropharyngeal abscess and surgical tracheostomy for airway protection. Unfortunately, although the procedure revealed purulent material, surgeons were unable to collect suitable samples for microbiological testing.
COMMENT: In figure 5, can you point the paramount reduction of inflammatory markers, as in Figure 4?
RESPONSE: Thank you for this important suggestion. We underline that, at day 9, the reduction of vasopressors was associated with the decrease of inflammatory markers. Thus, the values of IL-6 were added to the picture. We must underline that thanks to your comment, the figure gives a better idea of the effect of the therapy on the inflammatory cascade. (see the revised picture 5)
COMMENT: Did you check the SOFA scores daily? If so, can you put them in Table 1?
RESPONSE: We added it in table 1. There is evidence of a reduction of the SOFA score on day 9 with a reduction of inflammatory markers and improved hemodynamic support. This is further proof that the therapeutic choice was efficacy. Thanks for your suggestion.
COMMENT: Was follow up CT not done? Findings?
RESPONSE: Yes, it was done. We do not have explained what happened after discharge from ICU because our intention was to discuss the early diagnosis and treatment, just in the pro-inflammatory phase of the syndrome. After the discharge to rehabilitation care unit, the subject was in good health condition. A first follow up CT was done fifteen days after ICU discharge. It showed better lungs ventilation, absence of retropharyngeal abscess, laterocervical lymphadenopathy, and internal jugular thrombosis but persistent hepatosplenomegaly and some cavitary lungs lesions. Another thoracic CT scan was performed after three months. A summary of these features was added to the report.
COMMENT: The discussion is rather long.
RESPONSE: Although some aspects of the discussion are aimed at focusing the reader's attention on the key points of the report including the early diagnosis, source control, antibiotic therapy, anticoagulant therapy, and hemodynamic support, we have tried to improve the fluidity of the text.
COMMENT: Rather than describing each treatment options for Lemierre's syndrome, it is better to present treatment combinations (or cases) that have been reported to successfully treat Lemierre's syndrome. Is this the first report to treat Lemierre's syndrome with your treatment combination in the literature?
RESPONSE: Thank you. This is aim of this report. Although several aspects of the therapy are already known as cornerstone of the treatment of Lemierre’ syndrome, this submission underlines the triple drug synergism with emphasis on sequential therapeutics approaches (Fig 4). Notably, this is the first report that addresses the association norepinephrine, argipressin and steroid in this septic syndrome. We think that the knowledge of the early neuroendocrine dysregulation occurring in septic shock is the cornerstone of hemodynamic support and we emphasize the multimodal approach to limit the organ damage. This multimodal approach led to acid lactic values always under 2 mmol/l, despite the worsening of the radiological and laboratory data (Fig 6).
COMMENT: The conclusion is too vague; it should be more specific that reflects your specific study results. Can you only present how you treated this patient? Was anticoagulant therapy done in this patient?
RESPONSE: Thanks for your suggestion. From line 197 to 200 we explain that anticoagulant therapy was administered through the entire stay, and we discuss the use of anticoagulants in this syndrome according to literature data from line 292 to 299. On your suggestion, we added the drug enoxaparin in line 198.
Reviewer 3 Report
This manuscript is reporting a severe case of Lemierre's syndrome treated with source control, initiation of adequate antibiotic therapy, appropriate anticoagulant therapy, and hemodynamic optimization. The case appears of interest to the readership of Antibiotics, but a few of my concerns should be addressed before acceptance for publication to the journal.
1) Please revise your manuscript by having an English native speaker review your manuscript. General writing style in English should be improved substantially.
2) In your manuscript, a few antibiotics were listed as "susceptible" in your discussion. Did you perform any susceptibility testing? If so, what test did you use? Please clarify.
3) Lastly, it is my understanding that MIC susceptibility breakpoints are not available for Fusobacterium necrophorum. Can you address this aspect in your discussion..?
Author Response
COMMENT: Please revise your manuscript by having an English native speaker review your manuscript. General writing style in English should be improved substantially.
RESPONSE: Thanks for this suggestion. The test has been submitted to the attention of a native speaker.
COMMENT: In your manuscript, a few antibiotics were listed as "susceptible" in your discussion. Did you perform any susceptibility testing? If so, what test did you use? Please clarify.
RESPONSE: Unfortunately, in our laboratory of microbiology there were no cards for antibiotic susceptibility of anaerobic bacteria. So, when the microbiologist informed the isolation of F. necrophorum from blood cultures by Malditof spectrometry, we discussed about the best antibiotic choice according to literature data. The articles we referenced are the best performed and the more recent according to our opinion (ref 22-23-24). The therapy was already started with piperacillin-tazobactam in empiric therapy at admission in hospital; subsequently, in ICU, according to literature data of emerging isolation of F. necrophorum strains resistant to beta-lactamics (ref 25-26), the therapy was confirmed and implemented with metronidazole (ref 8).
COMMENT: Lastly, it is my understanding that MIC susceptibility breakpoints are not available for Fusobacterium necrophorum. Can you address this aspect in your discussion?
RESPONSE: The MIC susceptibility breakpoints of Fusobacterium were not disposable because there were not cards of susceptibility in our hospital for testing the susceptibility to antibiotics. The MIC breakpoints of gram negative anaerobes are available from the EUCAST clinical breakpoint tables version 11.0 (valid from 2021-01-01) in relation to MIC breakpoint for Gram negative-anaerobes and no breakpoints for susceptibility testing are given for cephalosporins, monobactams, fluoroquinolones, aminoglycoside, glycopeptides lypoglicopeptides , tetracyclines, oxazolidinones, macrolides; MIC breakpoints are instead available for clindamycin, penicillin, carbapenems , chloramphenicol and metronidazole. In absence of antibiogram we can’t refer to MIC breakpoints but just to literature data and clinical response.
About literature data, Jansen et al identified 357 isolated of F. necrophorum from human infections in Denmark . The isolated were analysed and the authors described their phenotypic characteristics .
Reviewer 4 Report
The authors present a case of Lemierre's syndrome and discussion of intensive care management. This syndrome is rare, but is well known to clinicians. There is nothing novel about the case that warrants publication in Antibiotics. Vasopressor management, source control, and antimicrobial therapy represent basic concepts of medicine. I don't see a particular aspect of management or discussion points presented that may change the practice of Antibiotics readers. This case report may be suitable for another journal.
Author Response
The authors present a case of Lemierre's syndrome and discussion of intensive care management. This syndrome is rare, but is well known to clinicians. There is nothing novel about the case that warrants publication in Antibiotics. Vasopressor management, source control, and antimicrobial therapy represent basic concepts of medicine. I don't see a particular aspect of management or discussion points presented that may change the practice of Antibiotics readers. This case report may be suitable for another journal.
Response:
We agree with the reviewer that many concepts related to the management of this syndrome and septic shock are well understood by the scientific community. However, we believe that the description of this clinical case has some peculiarities. Diagnostic guidance is often very complex, especially in contexts where not all the diagnostic tools are available. In our case, for example, a major limitation was the lack of cards for antibiotic susceptibility of anaerobic bacteria. Taken together, these gaps can compromise patient prognosis. Hence, a multidisciplinary/interdisciplinary approach is crucial for a good clinical outcome. Finally, we wanted to underline the particularities of this septic shock picture. In particular, the neuroendocrine dysregulation with vasoplegia can be devastating. Consequently, the maximization of oxygen delivery can be successfully performed through the administration of triple synergic therapy with norepinephrine, vasopressin, and hydrocortisone.
Obviously, these concepts are not a novelty in the management of septic shock, but we believe that they can be very interesting when contextualized to the Lemierre's syndrome. For these reasons, we suppose that this paper may be of interest to the reader. Notably, this is the first report that addresses the association norepinephrine, argipressin, and steroid in this type of septic syndrome.
Round 2
Reviewer 2 Report
- In Figure 1 legend, computed tomography and lung scan? or computed tomography of lung scan?
- In line 134, I think the grammar is wrong. I cannot understand the sentence.
- Others are were revised. Thank you for your hard work.
Author Response
QUESTION: In Figure 1 legend, computed tomography and lung scan? or computed tomography of lung scan?
- In line 134, I think the grammar is wrong. I cannot understand the sentence.
- Others are were revised. Thank you for your hard work.
RESPONSE: "Thank you for your evaluable work for improving the quality of this paper. We have really appreciated it."
Reviewer 4 Report
I do not see a significant improvement in the report to warrant changing prior recommendations. I am worried this report will encourage the use of combination therapy to treat anaerobic infections based on an N of 1. Can the authors find and cite the overall susceptibility rates of Fusobacterium in the literature? How often are these bacteria resistant to metronidazole? Why do we need combination therapy with piperacillin/tazobactam and metronidazole? Why do we promote the use of an antipseudomonal agent (piperacillin/tazobactam) for treatment of Fusobacterium? Aren't these practices detrimental to antimicrobial stewardship efforts?
Author Response
QUESTION: I do not see a significant improvement in the report to warrant changing prior recommendations. I am worried this report will encourage the use of combination therapy to treat anaerobic infections based on an N of 1. Can the authors find and cite the overall susceptibility rates of Fusobacterium in the literature? How often are these bacteria resistant to metronidazole? Why do we need combination therapy with piperacillin/tazobactam and metronidazole? Why do we promote the use of an antipseudomonal agent (piperacillin/tazobactam) for treatment of Fusobacterium? Aren't these practices detrimental to antimicrobial stewardship efforts?
RESPONSE: We have found 8 works of interest. To facilitate your analysis, we have prepared an ad hoc table that you can find at the end of this answer. In brief, it emerges that resistance is geographically variable. Moreover, all isolated were sensitive to metronidazole and emerging resistance to penicillin is evident in Europe whereas clyndamicin resistance is major in Asia.
In accordance with your suggestions, we have deemed it appropriate to make some clarifications in the paper. They concern the geographical contextualization of the problem of the antimicrobial resistance. In particular, pip/tazo is considered a weak inducers of beta-lactamases such as AmpC and this is why it is the most used empiric beta-lactam in Italy, not for the anti-Pseudomonas activity (Akata K, Muratani T, Yatera K, Naito K, Noguchi S, Yamasaki K, Kawanami T, Kido T, Mukae H. Induction of plasmid-mediated AmpC β-lactamase DHA-1 by piperacillin/tazobactam and other β-lactams in Enterobacteriaceae. PLoS One. 2019 Jul 8;14(7):e0218589. doi: 10.1371/journal.pone.0218589. PMID: 31283769; PMCID: PMC6613692.). Finally, pip/tazo is used in Italy for its activity against ESBL gram negative bacteria according to a carbapenem-sparing strategy to avoid a stronger selection of MDR microorganism in the microbiota of treated patients. Because Italy is an ESBL-country differently from countries of northern Europe, pip/tazo is preferred to ampi/sulba, although the anti-anaerobic activity is similar.
We hope you can appreciate our work.
Best regards
|
Authors |
N. of isolates |
Species |
Year |
Nation |
Penicillin R |
Amoxicilin-ac.clavulanic R (%) |
Ampicillin R (%) |
Amoxicillin R (%) |
Clyndamicin R (%) |
Cefoxitin R (%) |
Cefotetan R (8%) |
Metronidazole R (%) |
Meropenem R (%) |
Imipenem R (%) |
Piperacillina-tazobactam R (%) |
Moxifloxacin R (%) |
Chloramphenicol R (%) |
||
|
Austin [i] |
149 |
Fusobacterium spp |
2010-11 |
Ontario |
12.1% |
- |
- |
- |
6.1% |
9.1% |
|
0% |
0% |
- |
0% |
0 % |
- |
|
|
|
Kim [ii] |
21 |
Fusobact necrophorum |
2003-2020 |
Korea |
0 % |
- |
- |
- |
0 % |
0 % |
|
0 % |
- |
0 % |
0 % |
0 % |
0 % |
|
|
|
Shilnikova[iii] |
13 |
Fusobacterium spp |
2004-2014 |
Russia |
0 % |
0 % |
- |
- |
0 % |
- |
|
0 5 |
0 % |
0 % |
- |
- |
- |
|
|
|
Byun [iv] |
19 |
Fusobacterium spp |
2014-2016 |
Korea |
16% |
- |
- |
- |
21% |
0% |
0% |
0% |
0% |
0% |
- |
11% |
0% |
|
|
|
Maraki [v] |
30 |
Fusobacterium spp |
2016-2019 |
Greece |
6.7 % |
3.3% |
6.7% |
- |
6.7% |
3.3% |
- |
0% |
0% |
3.3% |
3.3% |
26.7% |
0% |
|
|
|
Yusuf[vi] |
230 |
Fusobacterium spp |
2004-2014 |
Belgium |
2% |
1% |
- |
- |
1% |
- |
- |
0% |
- |
- |
- |
- |
-
|
|
|
|
Pintor[vii] |
22 |
Fusobacterium spp |
2018-2019 |
Spain |
5% |
0% |
- |
- |
10% |
- |
- |
- |
- |
5% |
- |
- |
- |
|
|
|
Veloo[viii] |
39 |
Fusobacterium spp |
2011-2013 |
Netherlands |
- |
5% |
- |
15% |
0% |
- |
- |
0% |
- |
- |
- |
- |
- |
|
|
[i] Marchand-Austin A, Rawte P, Toye B, Jamieson FB, Farrell DJ, Patel SN. Antimicrobial susceptibility of clinical isolates of anaerobic bacteria in Ontario, 2010-2011. Anaerobe. 2014 Aug;28:120-5. doi: 10.1016/j.anaerobe.2014.05.015. Epub 2014 Jun 9. PMID: 24923267.
[ii] Kim M, Yun SY, Lee Y, Lee H, Yong D, Lee K. Clinical Differences in Patients Infected with Fusobacterium and Antimicrobial Susceptibility of Fusobacterium Isolates Recovered at a Tertiary-Care Hospital in Korea. Ann Lab Med. 2022 Mar 1;42(2):188-195. doi: 10.3343/alm.2022.42.2.188. PMID: 34635612; PMCID: PMC8548237.
[iii] Shilnikova II, Dmitrieva NV. Evaluation of antibiotic susceptibility of Bacteroides, Prevotella and Fusobacterium species isolated from patients of the N. N. Blokhin Cancer Research Center, Moscow, Russia. Anaerobe. 2015 Feb;31:15-8. doi: 10.1016/j.anaerobe.2014.08.003. Epub 2014 Aug 23. PMID: 25157873.
[iv] Byun JH, Kim M, Lee Y, Lee K, Chong Y. Antimicrobial Susceptibility Patterns of Anaerobic Bacterial Clinical Isolates From 2014 to 2016, Including Recently Named or Renamed Species. Ann Lab Med. 2019 Mar;39(2):190-199. doi: 10.3343/alm.2019.39.2.190. PMID: 30430782; PMCID: PMC6240532.
[v] Maraki S, Mavromanolaki VE, Stafylaki D, Kasimati A. Surveillance of antimicrobial resistance in recent clinical isolates of Gram-negative anaerobic bacteria in a Greek University Hospital. Anaerobe. 2020 Apr;62:102173. doi: 10.1016/j.anaerobe.2020.102173. Epub 2020 Feb 7. PMID: 32062399.
[vi] Yusuf E, Halewyck S, Wybo I, Piérard D, Gordts F. Fusobacterium necrophorum and other Fusobacterium spp. isolated from head and neck infections: A 10-year epidemiology study in an academic hospital. Anaerobe. 2015 Aug;34:120-4. doi: 10.1016/j.anaerobe.2015.05.006. Epub 2015 May 16. PMID: 25988544.
[vii] López-Pintor JM, García-Fernández S, Ponce-Alonso M, Sánchez-Díaz AM, Ruiz-Garbajosa P, Morosini MI, Cantón R. Etiology and antimicrobial susceptibility profiles of anaerobic bacteria isolated from clinical samples in a university hospital in Madrid, Spain. Anaerobe. 2021 Sep 11;72:102446. doi: 10.1016/j.anaerobe.2021.102446. Epub ahead of print. PMID: 34520862.
[viii] Veloo AC, van Winkelhoff AJ. Antibiotic susceptibility profiles of anaerobic pathogens in The Netherlands. Anaerobe. 2015 Feb;31:19-24. doi: 10.1016/j.anaerobe.2014.08.011. Epub 2014 Sep 1. PMID: 25192966.